# Variation in methods, results and reporting in electronic health record-based studies evaluating routine care in gout: A systematic review

**Samantha S. R. Crossfield**[1]*, **Lana Yin Hui Lai**[1], **Sarah R. Kingsbury**[1,2], **Paul Baxter**[3], **Owen Johnson**[4], **Philip G. Conaghan**[1,2‡], **Mar Pujades-Rodriguez**[5‡]

**1** Leeds Institute of Rheumatic and Musculoskeletal Medicine, University of Leeds, Leeds, United Kingdom, **2** NIHR Leeds Biomedical Research Centre, Leeds, United Kingdom, **3** Leeds Institute of Cardiovascular and Metabolic Medicine, University of Leeds, Leeds, United Kingdom, **4** School of Computing, University of Leeds, Leeds, United Kingdom, **5** Leeds Institute of Health Sciences, University of Leeds, Leeds, United Kingdom

‡ These authors are joint senior authors on this work.
* s.crossfield@leeds.ac.uk

**Data Availability Statement:** All relevant data are within the manuscript and its Supporting

## Abstract

### Objective

To perform a systematic review examining the variation in methods, results, reporting and risk of bias in electronic health record (EHR)-based studies evaluating management of a common musculoskeletal disease, gout.

### Methods

Two reviewers systematically searched MEDLINE, Scopus, Web of Science, CINAHL, PubMed, EMBASE and Google Scholar for all EHR-based studies published by February 2019 investigating gout pharmacological treatment. Information was extracted on study design, eligibility criteria, definitions, medication usage, effectiveness and safety data, comprehensiveness of reporting (RECORD), and Cochrane risk of bias (registered PROSPERO CRD42017065195).

### Results

We screened 5,603 titles/abstracts, 613 full-texts and selected 75 studies including 1.9M gout patients. Gout diagnosis was defined in 26 ways across the studies, most commonly using a single diagnostic code (n = 31, 41.3%). 48.4% did not specify a disease-free period before 'incident' diagnosis. Medication use was suboptimal and varied with disease definition while results regarding effectiveness and safety were broadly similar across studies despite variability in inclusion criteria. Comprehensiveness of reporting was variable, ranging from 73% (55/75) appropriately discussing the limitations of EHR data use, to 5% (4/75) reporting on key data cleaning steps. Risk of bias was generally low.

Information files. All data was collected from the referenced previously published manuscripts.

**Funding:** SSRC was supported in this work by the Medical Research Council (MRC) Leeds Medical Bioinformatics Centre (MR/L01629X). SRK and PGC are funded in part by the National Institute for Health Research (NIHR) through the NIHR Leeds Biomedical Research Centre and the Versus Arthritis Experimental Osteoarthritis Treatment Centre (20083). The views expressed are those of the authors and not necessarily those of the NHS, the NIHR or the Department of Health. The funders had no role in study design, data collection and analysis, decision to publish, or preparation of the manuscript. URLs for the funder websites: https://lida.leeds.ac.uk/research/mbc/, https://app.dimensions.ai/details/grant/grant.5136386.

**Competing interests:** The authors have declared that no competing interests exist.

## Conclusion

The wide variation in case definitions and medication-related analysis among EHR-based studies has implications for reported medication use. This is amplified by variable reporting comprehensiveness and the limited consideration of EHR-relevant biases (e.g. data adequacy) in study assessment tools. We recommend accounting for these biases and performing a sensitivity analysis on case definitions, and suggest changes to assessment tools to foster this.

## Introduction

A growing number of health organizations routinely use electronic health record (EHR) systems for patient management [1]. Many systems include electronic prescribing and these records enable evaluation of patient management and exploration of issues such as guideline-indicated treatment and medication adherence. This is informative for diseases that are primarily managed with pharmacological interventions.

Musculoskeletal diseases are common in our ageing and increasingly obese populations. Gout is the most common inflammatory musculoskeletal disease, with prevalence estimates ranging between 0.1 and 10% worldwide [2]. It results from monosodium urate (MSU) crystal deposition in articular and peri-articular tissues that leads to debilitating flares and joint damage [3]. The management of gout involves medication usage for both acute and chronic cases [3–5].

The growing coverage of EHRs, including electronic prescribing, has led to increasing numbers of studies being conducted using this type of data to assess the burden and management of chronic diseases in real-world settings. EHRs have been used in gout research to examine temporal and demographic variations in treatment, quality of management and patient outcomes [6–9]. However, EHRs are primarily designed to facilitate continuity and billing in healthcare provision. Secondary use for research requires understanding of how EHRs are used in clinical practice in order to design studies appropriately (e.g. clinicians may use different codes to record an event depending on their training, experience and the design of the EHR system). EHR-based studies use a variety of approaches to define and validate events or cases, ascertain medication use and outcomes, and report methods and findings. An understanding of the main factors that determine heterogeneity in estimates is essential for interpreting study findings. Variation in approaches to EHR-based research and the impact that method selection might have on results, as well as variation in comprehensiveness of reporting and study quality of those studies, have not previously been assessed.

We therefore aimed to systematically review methods, results, reporting and risk of bias in EHR-based studies evaluating pharmacological management. Gout, being a common rheumatologic condition with predominant pharmacological management, and the focus of several published EHR-based studies, was selected as an exemplar for this evaluation.

## Materials and methods

### Search strategy and study selection

We registered the study protocol in the International Prospective Register of Systematic Reviews (PROSPERO), number CRD42017065195 [10]. The systematic review was conducted in line with the Preferred Reporting Items for Systematic Reviews and Meta-Analyses

(PRISMA) statement (S1 Table) and the recommendations of Denison et al. [11, 12]. We searched Scopus, Web of Science, CINAHL, PubMed, MEDLINE, EMBASE and Google Scholar on 20 February 2019 for papers published since 1 January 1970 (Supplementary Methods in S1 File). The search strategy combined search terms for 'electronic health records', 'gout' and 'medication' with their synonyms (S2 Table). Two reviewers (SSRC and LLYH) hand-searched relevant reviews, conferences, meeting and protocol publications and screened titles and abstracts using Rayyan [13]. Full texts were then screened to select all EHR population-based studies of gout, reporting on treatment utilization, effectiveness or safety (selection criteria listed in Supplementary Methods in S1 File). A third reviewer (MP-R or SK) resolved discrepancies.

## Data extraction and quality assessment

A data extraction form was developed using Microsoft Access 2013, based on the recommendations of the Centre for Reviews and Dissemination for general information, study and participant characteristics, setting and results [14]. Characteristics such as 'recruitment procedures, costs and resource use' were adapted to EHR-relevant details. Clinical and epidemiological guidance from PC, PB and MP-R informed the data elements extracted. We classified the approach taken to identify gout diagnoses as "liberal" if there was a risk of over-classifying (high sensitivity) and "specific" if there was risk of under-classifying. Liberal approaches required a single diagnostic code, free-text keyword or prescription unless it was recorded by a rheumatologist. Specific approaches used further requirements concerning specialist care setting, having further diagnostic codes or prescriptions, having tests or meeting guideline diagnostic criteria such as the 2015 American College of Rheumatology (ACR) / European League Against Rheumatism criteria [15]. Two investigators (SSRC and LLYH) independently abstracted the data and reviewer discrepancies were resolved by consensus with a third reviewer (MP-R).

Comprehensiveness of reporting (CoR) of EHR data use was assessed using the REporting of studies Conducted using Observational Routinely-collected Data (RECORD) statement [16]. Study quality was assessed using the Cochrane Tool to Assess Risk of Bias (RoB) in Cohort Studies [17]. CoR and RoB scores were calculated applying the Care Quality Commission Survey Scoring Method; answers ranked from 0–10 for the least to most positive answer option [18]. The sum of answers per study was divided by the number of questions evaluated to obtain an overall score for each study: 'perfect' scores were 10/10 CoR and 15/15 RoB. Where a question was 'not applicable', it was excluded and did not affect the score [19].

## Study outcomes

The outcomes were: indicators of gout diagnosis; medication types considered; methods/ results relating to treatment utilization (including their period of assessment in relation to the timing of gout diagnosis); effectiveness and safety and association between these results and the gout definition used (liberal or specific); CoR on EHR data use and RoB indicators, including analyses of time-trend and according to gout definition and study size.

## Statistical analysis

We performed descriptive statistics, using Microsoft SQL 2014 and Excel 2013, to describe definitions of gout, methods used, treatment utilization, effectiveness and safety, reporting quality and RoB. To account for study sample size, weighted proportions and means/medians were calculated for overall estimates of medication use and treatment outcomes. Change in scores

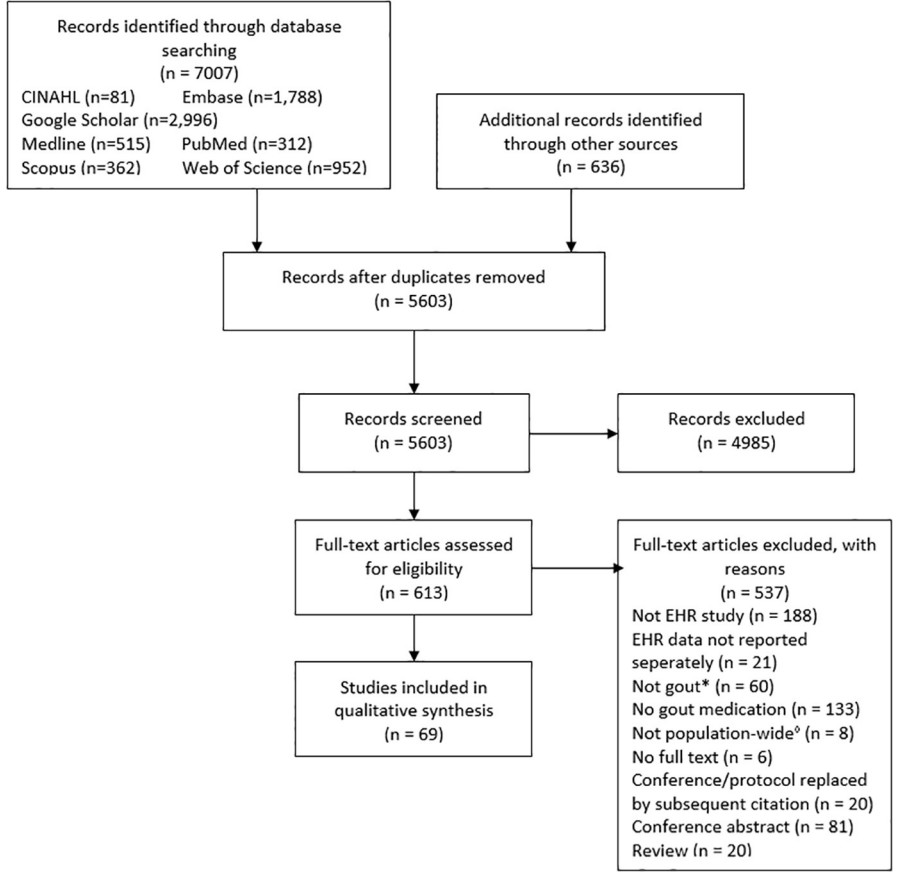

**Fig 1. Flow chart of study identification and selection.** CINAHL, Cumulative Index of Nursing and Allied Health Literature; EHR, electronic health record. * Studies including asymptomatic hyperuriceamia. ◇Studies using databases that are restricted to specific (non-gout) sub-populations (e.g. an adverse event database).

based on cohort size and following the publication of CoR guidelines were assessed using Chi-square tests for a linear trend.

## Results

### Study and cohort characteristics

Titles/abstracts for 5,603 articles were screened, 613 full-text articles were reviewed and 75 met the eligibility criteria (Fig 1, S3 Table). All studies were published between 2002 and 2019, with a rising publication rate ($R^2$ = 0.86) (S1 Fig). Most were conducted in the UK (n = 26) (Table 1). Amongst 67 studies that reported the calendar period covered, the median duration was 8 years (interquartile range (IQR) 3.5–15) and 52 (77.6%) used data recorded since 2010. Fifty (66.7%) analysed data from primary care. Thirty-eight studies (50.7%) reported the number of sites considered and 19 (50.0%) of them were single-center (range 1–15 520). Only 31 (41.3%) reported the population size from which the cohort was drawn (range 8686–35 million).

Besides inclusion criteria regarding gout diagnosis or medication, some studies used further criteria regarding: comorbidity (n = 17, 22.7%), age (n = 38, 50.7%), minimum period of enrollment before follow-up start (n = 22, 29.3%), minimum follow-up duration (n = 14, 18.7%), minimum number of visits during the study or current registration status (n = 13,

**Table 1. Characteristics of studies included (n = 75).**

| Characteristic | n (%) | Characteristic | n (%) |
|---|---|---|---|
| **Geographic Setting*** | | **Site Type** | |
| Western Europe | 41 | Primary care | 29 (39) |
| North America | 25 | Primary care and hospital | 21 (28) |
| Asia | 8 | Hospital | 13 (17) |
| Australia / New Zealand | 4 | Outpatient | 4 (5) |
| Middle East | 1 | National dataset | 7 (9) |
| Not specified | 2 | Nursing Facility | 1 (1) |
| **Study Design** | | **Year of Publication** | |
| Site-randomized trial (usual care cohort) | 1 (1) | 2000–2004 | 1 (1) |
| Matched cohort | 6 (8) | 2005–2009 | 6 (9) |
| Cohort | 46 (61) | 2010–2014 | 24 (35) |
| Case Control | 16 (21) | 2015–February 2019 | 44 (64) |
| Cross-sectional | 6 (8) | **Gout Cohort Size** | |
| **Study Aim** | | ≤100 | 7 (9) |
| Epidemiology of gout | 22 (29) | 101–1,000 | 21 (28) |
| Patient management | 6 (8) | 1,001–10,000 | 15 (20) |
| Adherence to clinical guidelines | 12 (16) | 10,001–100,000 | 22 (29) |
| Adherence and gaps in therapy | 5 (7) | >100,000 | 7 (9) |
| Treatment safety | 10 (13) | Not specified | 3 (4) |
| Treatment effectiveness | 3 (4) | | |
| Patient knowledge, beliefs & education | 1 (1) | | |
| Epidemiology; patient management | 7 (9) | | |
| Other combination | 9 (12) | | |

*Some studies had multiple applicable settings.

17.3%). Six studies (8.0%) excluded patients with missing demographic, prescription or laboratory values and 8 (10.7%) only included sites that met data entry quality standards. While 68 (90.7%) selected all eligible patients, 4 (5.3%) selected a random sample, and 3 (4.0%) consenting patients. Gout cohort sizes increased over time (median 4368 patients, IQR 435–30 767). Fifty-nine (78.7%) reported the sex distribution, 50 (66.7%) the mean/median age and 11 (14.7%) socio-economic status. Twenty-five (31.9%) reported mean/median follow-up duration, which was over 5 years for 12 studies.

## Gout definition

Of 66 (88.0%) studies specifying the gout definition, 38 applied a liberal and 28 a specific approach; 58 (87.9%) used diagnostic codes, 13 (19.7%) medication, 6 (9.1%) test results and 3 (4.7%) free text (Tables 2 and 3). Diagnostic coding was optional in 8 studies and required by 50 (of which 12 also required additional criteria). When using diagnostic codes, 51 (87.9%) referenced the coding system and 25 (43.1%) provided the code-list. None provided the medication code-list (7/75 (9.3%) provided these for medication variables). Three (23.1%) studies had the applicable window of medication exposure before the study period, 5 (38.5%) during, 1 (7.7%) before or during [20], 1 (7.7%) current [21] and 3 (23.1%) did not specify it. Eleven (14.7%) repeated their analyses using different gout definitions (sensitivity analysis). Thirty-one (41.3%) defined incident gout: 29 (93.5%) by the first coded appearance and 16 (51.6%) required a prior 1–5 years with no diagnosis or medication.

**Table 2. Definitions of gout and medication exposure (n = 75).**

| Definitions | | n (%)* |
|---|---|---|
| **Gout Definition** | | |
| ≥1 diagnosis | | 31 (45) |
| ≥1 EHR reference (not specified) | | 2 (3) |
| ≥1 gout medication prescription/dispense | | 2 (3) |
| ≥1 diagnosis or gout medication | | 2 (3) |
| ≥1 diagnosis or keyword | | 2 (3) |
| ≥1 keyword search of EHR | | 1 (1) |
| ≥1 diagnosis; 1 diagnosis and medication (2 definitions) | | 4 (5) |
| 1 liberal and ≥1 specific definition (other than above) | | 4 (5) |
| ≥2 diagnoses | | 3 (4) |
| Survey response and ≥1 diagnosis | | 2 (3) |
| ≥1 diagnosis or medication and coded CKD, urolithiasis, tophus or >2 flares | | 2 (3) |
| ≥ 1 test | | 2 (3) |
| Meet ACR criteria | | 3 (3) |
| Other specific definition/s (not seen in >1 study) | | 9 (12) |
| No definition given | | 6 (8) |
| **Incident Gout Definition** | | 33 (44) |
| First code in the study or EHR | (% of 33) | 31 (94) |
| No diagnosis in prior time period (1–3 y) | (% of 33) | 13 (39) |
| Distinct codes for incident and prevalent | (% of 33) | 1 (3) |
| No diagnosis and/or medication in prior time period | (% of 33) | 5 (15) |
| No definition given | (% of 33) | 2 (6) |
| **Medication Minimum Duration/Dose Requirement** | | 8 (11) |
| Minimum of 6 months | | 4 (5) |
| Minimum of 3 consecutive months | | 1 (1) |
| Minimum of 1 month | | 1 (1) |
| Minimum of 2 prescriptions | | 1 (1) |
| ≥300mg/day of allopurinol | | 1 (1) |
| **Medication Exposure Measure** | | |
| Binary 'ever exposed' at any point in the study | | 23 (31) |
| Binary 'ever exposed' at a specific time point | | 14 (19) |
| Binary 'ever exposed' in a specific time window | | 9 (12) |
| Exposure within a window | | 26 (35) |
| Continuous exposure | | 4 (5) |
| Cumulative exposure | | 3 (4) |
| **Reporting on Medication Exposure** | | |
| Use at baseline or prior to study | | 35 (47) |
| Dosage | | 33 (44) |
| % 'ever exposed' during the study | | 29 (39) |
| Use at or during follow-up periods | | 19 (25) |
| Temporal duration of medication use | | 9 (12) |
| Use in chronological periods | | 8 (11) |

CKD, chronic kidney disease.

*Percentage is given as n out of 75 unless otherwise specified.

**Table 3. Distribution of studies according to elements considered in the definition of gout and medication exposure and their classification recording system (n = 75).**

| Indicator | | Count (%) |
|---|---|---|
| **Gout Diagnosis** | | |
| **Diagnostic Code** | | 58 (88) |
| *With provision of code-list* | (% of 58) | 25 (43) |
| ICD | (% of 58) | 33 (57) |
| Read Code / Oxmis | (% of 58) | 18 (31) |
| *Classification not specified* | (% of 58) | 7 (12) |
| **Medication** | | 13 (20) |
| *With provision of code-list* | (% of 13) | 0 (0) |
| Multilex | (% of 13) | 5 (39) |
| BNF | (% of 13) | 1 (8) |
| *Classification not specified* | (% of 13) | 7 (54) |
| **Test Result**[*] | | 6 (9) |
| UA crystals in synovial fluid | | 4 |
| Radiologic evidence, e.g. DECT scan | | 2 |
| Biopsy of tophus or synovial tissue | | 1 |
| High SUA level | | 2 |
| **Free text** | | 3 (5) |
| **Medication Exposure** | | |
| **Medication** | | 75 (100) |
| Multilex | | 7 (9) |
| ATC | | 7 (9) |
| National ID | | 2 (3) |
| BNF | | 1 (1) |
| *Classification not specified* | | 58 (77) |

ATC, Anatomical Therapeutic Chemical; BNF, British National Formulary; DECT, dual-energy computed tomography; ICD, International Classification of Diseases; SUA, serum uric acid; UA, uric acid

[*]Some studies used multiple tests in defining gout.

## Medication assessment

Forty-five (60.0%) studies used prescriptions, 20 (26.7%) dispensary data and 3 (4.0%) both (S1 Chart). Allopurinol, non-steroidal anti-inflammatories (NSAIDs) and colchicine were the most common of 27 gout-related drugs or groups reported (S4 Table). Eight studies specified a minimum prescription duration or dose.

## Measures of medication use

Most studies reported the percentage with medication 'ever used' (n = 44, 59%, of which 26 only reported this) or use within specified windows (n = 26, 34.7%). Seven (9.3%) reported continuous or cumulative exposure, 8 (10.7%) temporal prescribing trends and 32 (42.7%) the prescribed dosage. Sixteen (21.3%) assessed the proportion initiating treatment at diagnosis or in periods of follow-up, 2 prescription gaps and 1 the percentage with ≥60 consecutive days of prescribing. Urate-lowering therapy (ULT) adherence was measured as a medication possession ratio (MPR) or the proportion of days covered (PDC) ≥0.80 in 1 and 8 (10.7%) studies respectively.

## Reported estimates of medication use

Medication initiation was low although higher in studies with stricter definitions, with little temporal change (S1 Chart). Studies selecting incident patients with ≥1 gout diagnosis (i.e. ≥1 diagnostic code) reported 6.7% (range 0–9.4%) of patients initiating ULT at diagnosis and 22.9% (range 16.9–25.4%) by 12 months [22, 23]. In studies also requiring prior registration without diagnosis, this was 15% at diagnosis and 27.7% (range 23–31.9%) by 12 months [24–28]. Kapetanovic et al. reported 47.8% and 60.6% of patients with incident gout receiving ULT in 2011 when using a liberal and specific definition respectively [29]. Studies using a liberal definition found stable ULT prescribing (mean 28.3%), declining NSAID use (mean 36.3%) and rising colchicine use (mean 6.3%) across 1990 to 2014 among patients with incident and prevalent gout [24, 25, 27, 30–32]. Arromdee et al. used a specific definition and reported higher colchicine use: 19.8% in 1995–1996 [33].

ULT duration was short although longer in studies with a specific gout definition. It ranged from 0.33–0.8 years in studies requiring ≥1 diagnosis (liberal) and 2.5–4.0 years in studies including patients with ≥2 diagnoses [23, 34–37]. Scheepers et al. reported mean PDC of 0.57 (standard deviation (SD) ±0.34) for patients with ≥1 diagnosis, while Coburn et al. selected patients with ≥2 diagnoses and reported median PDC 0.7 to 0.83 [23, 38]. The proportion with PDC≥80 in the first year was 38.6% among patients with ≥1 diagnosis [27, 39] and 59% among patients with ≥1 diagnosis validated through a survey [40]. MPR was only measured using a liberal definition [41].

ULT doses were low with limited up-titration although greater in studies with a specific gout definition. The mean proportion of patients with ULT up-titration was 5.4% (range 4–36%) and 29.0% (range 22.4–39.3%) when using liberal and specific definitions respectively [35, 37, 41–44]. The mean initial allopurinol dose was 148.1 mg/day and overall dose was 223.3 mg/day: 194.1 mg/day and 231.4 mg/day in studies with liberal and specific definitions respectively [21, 34, 42, 45–50]. Inappropriate allopurinol dosing for renal disease patients was high (mean 24.8%, range 22–25.9%) and only reported in studies using liberal definitions [51, 52].

## Measures of treatment outcomes

Nine (12.0%) studies measured changes in serum urate (SUA) level with ULT: 7 measured mean change and 4 the percentage achieving SUA level goal. Eight (10.7%) examined the impact of ULT on disease control or SUA level, with 5 assessing associations with the starting dose, titration and drug adherence. Other measures evaluated were the percentage of patients reaching the SUA goal or switching treatment, the mean SUA change per treatment group, comparison of changes in repeated-measures and time to response. Fifteen (20%) evaluated treatment safety, with 9 determining effect on the risk of fracture, joint replacement, mortality, myopathy, chronic kidney disease, hepatoxicity and cardiovascular disease.

## Reported estimates of treatment outcomes

Studies, especially those with specific gout definitions, reported higher SUA level goal attainment with higher starting allopurinol dose, adherence and up-titration [35, 38–40, 47]. For example, Mantarro et al. used a liberal definition and reported lower odds of hyperuriceamia in the first 90–149 days for adherent patients (AOR, 0.40; 95%CI, 0.24–0.67) [39]. Rashid et al., with a specific definition, reported higher odds of goal attainment associating with higher starting doses (AOR, 1.92; 95% CI, 1.86–2.22 for 100–300 mg compared with ≤100mg) and adherence (AOR, 2.52; 95% CI 2.41–3.01) [35]. ULT had a positive dosage- and duration-dependent effect on SUA level (7/7 studies), and combination therapy was more effective than monotherapy (2/2 studies), regardless of gout definition [35, 38–40, 44, 46, 47, 49, 53].

Most studies reported that treatment was safe and well tolerated, with few switches, regardless of gout definition. For example, with a liberal definition, colchicine associated with lower cardiovascular risk and mortality compared to no medication; and with a specific definition it was unrelated to myopathy risk when prescribed with vs without statin [54–56]. Only the association between allopurinol and all-cause mortality was examined using varied definitions (both used propensity-matched cohorts): Kuo et al. used ≥1 diagnosis and reported no improvement compared with non-exposed gout patients (HR, 1.01; 95% CI, 0.92–1.09); Coburn et al. used ≥2 diagnostic codes and noted no improvement with titration (HR, 1.08; 95% CI, 1.01–1.17) [38, 57]. Both were hindered in investigating dosage-dependent effects by pervasive low dosage prescribing.

## Comprehensiveness of reporting

The overall mean score for CoR of EHR data use was 5.2/10 (SD±1.5): 5.6/10 (SD±1.3), 5.3/10 (SD±1.4) and 3.0/10 (SD±0.9) for studies with a liberal, specific and no stated approach to defining gout respectively (range 1.8–8.5). Those with a liberal definition reported less comprehensively on validation, database population and linkage methodology but more frequently provided full code-lists and the patient count at each selection stage.

In the title/abstract, 66 (88.0%) mention ("yes"/"partly") the data (although 11 only name the database) while 15 (20.0%) reported the geographic region, 9 (12.0%) the study timeframe and 32 (42.7%) both (Fig 2, S5 Table). In the methods, cohort selection was the most comprehensively recorded ("yes"/"partly" = 70, 93.3%), while only 7 (9.3%) provided the codes or algorithms for all variables, 6 (8.0%) provided a data availability statement and 4 (5.3%) made ≥2 references to data cleaning or preparation. Seventeen (22.7%) provided incorrect references to a gout diagnosis validation study; 7 referenced a study by Meier et al. that used more specific selection criteria [58], 5 referenced studies using a different coding system to that used in the study and 3 referenced studies that validated a disease other than gout. Fifteen used data from the Clinical Practice Research Datalink, which trains clinicians in data-entry [59], but only 8 (53.3%) noted this.

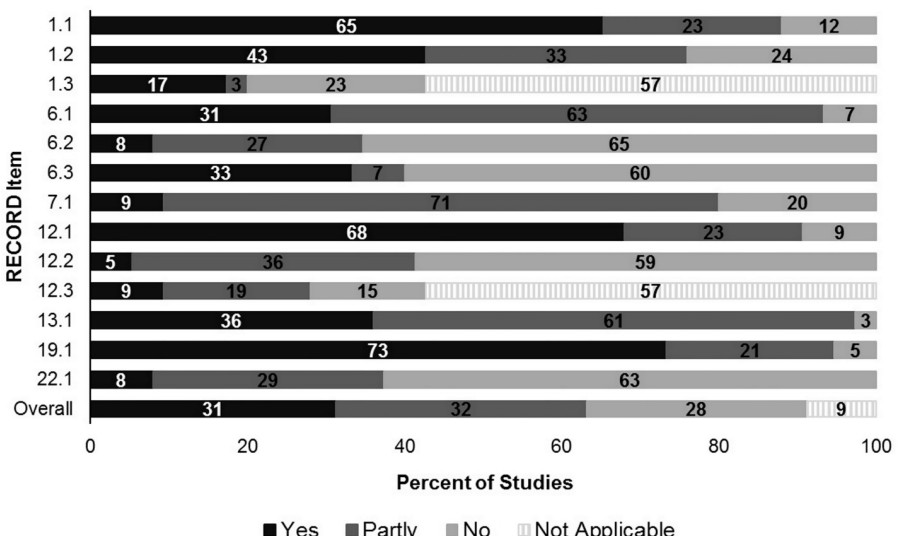

**Fig 2. Percentage of studies with comprehensive reporting on RECORD items (n = 75).** RECORD, REporting of studies Conducted using Observational Routinely-collected Data.

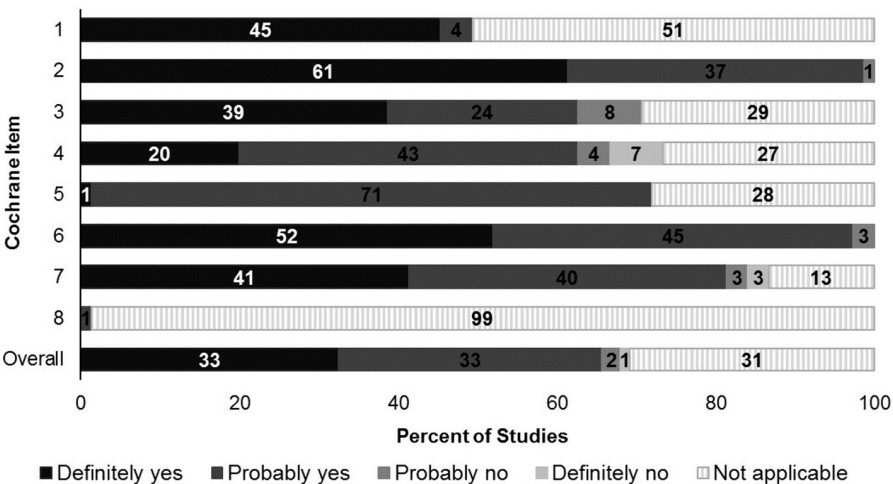

**Fig 3. Percentage of studies with low risk of bias as assessed with the Cochrane Tool for Cohort Studies (n = 75).**

Overall CoR scores rose over time (S2 Fig) and were higher after 2015, when RECORD guidelines were published ($x^2 = 5.7$, *p* = 0.02). The median score was 5.0 (IQR = 3.6–5.9) for 40 papers published by 2015, and 5.8 (IQR = 5.0–6.5) from 2016 (n = 35). The proportion scoring "yes" or "partly" was greater for all items from 2016 except 13.1, with a mean change of +15.5% (SD ±16.1). This ranged from 0.4% decrease in studies describing the selection of included persons, to 56.3% increase in those describing the linkage methodology. However, from 2016, "yes" scores decreased for items 1.1, 6.1, 6.3, 19.1; notably 15.4% fewer mentioning the data type in the abstract and 14.3% fewer discussing the limitations of secondary data use. The proportion with "yes" scores remained ≤25% for 6/13 items (6.2–7.1, 12.2, 12.3, 22.1).

### Risk of bias

The mean RoB score was 12.0/15 (SD±1.6, range 7.5–15) and, where items were applicable to a study, >85% of studies were scored with low or probably low RoB for each item. All studies scored "yes", "probably yes" or "not applicable" for having exposed and non-exposed cohorts drawn from the same population and confidence in the assessment of the presence or absence of prognostic factors (Fig 3). In 8 (10.7%) studies patients were inappropriately matched or estimates were incorrectly adjusted and 6 (8.0%) inadequately assessed the outcome at follow-up start. However, the RoB measures were not applicable to, on average, 30.8% of studies per item (IQR 10.0–34.7%). The similarity of co-interventions between groups compared, and whether cohorts were drawn from the same population, were non-applicable for 74 (98.7%) and 38 (50.7%) studies respectively.

Mean RoB scores were 12.0 (SD±1.3), 11.8 (SD±1.8) and 12.6 (SD±1.9) for studies with a liberal, specific and unspecified gout definition. Overall RoB scores did not change over time ($x^2 = 2.8$, *p* = 0.09), (S3 Fig) and were similar amongst studies with <1000 and ≥1000 gout patients ($x^2 = 0.38$, *p* = 0.53) (S4 Fig).

### Discussion

This is the first systematic literature review of variation in methods, results and reporting in EHR-based studies of gout medication. The studies demonstrated wide variation in definitions and medication-related methods. This did not widely affect effectiveness and safety estimates,

though reporting quality was variable. Risk of bias was acceptable as assessed by the Cochrane tool although its applicability to EHR-based studies was limited.

## Gout definition

Variation in gout definition and medication-related methods may explain differences in medication utilization estimates across the studies. Most studies employed liberal approaches that may lead to more misdiagnosis (false positives) when suspected cases are recorded in EHRs during pre-diagnostic evaluation. Specific approaches are likely to have higher specificity but may lack sensitivity for identifying mild or recently diagnosed cases (i.e. severe or long-standing cases benefit from additional clinical contact, increasing opportunities for prescribing, tests and diagnostic coding). We recommend that EHR-based studies replicate key analyses using a liberal and specific definition (sensitivity analysis) and discuss potential implications of any discrepancies. We also advise that diagnostic definitions adopt contemporary guideline diagnostic criteria as far as possible and describe any constraints encountered in doing so.

## Medication assessment

No studies considered time since diagnosis, and assessment of severity or flare frequency is difficult in EHR data [36]. The medications studied were varied and little was reported regarding cumulative dosage and timing in relation to clinical events such as testing. Most studies measured variables that are well recorded in EHRs: SUA level, medication, comorbidities and procedures. There were varied approaches to measuring adherence and we recommend the practice of accounting for dose changes and early fills to avoid over-estimation.

## Medication use and treatment outcomes

EHR-based research has used prescription data to reflect the real-world challenges faced by clinicians in management of gout. ULT initiation, adherence and titration was sub-optimal and doses rarely reached above 300 mg/day. The studies in this review used EHR data from 1977 to 2017 with comprehensive coverage of prescriptions and consistently found a lack of improvement in gout pharmacological management over time. This is problematic given the disassociation from guideline recommendations and trial evidence showing that doses ≤300 mg/day fail to reduce SUA levels [4]. Estimates of medication use were higher and less heterogeneous in studies using specific gout definitions. These studies may select more severe and long-standing cases, so gout management may be slightly more appropriate in these instances. If prescribing improves, EHR-based research could evaluate prescribing patterns, dosage-dependent safety and effectiveness, and safety of concomitant prescribing. ULT was generally reported as safe and effective regardless of definition or methodology, which indicates the opportunity for optimizing gout control though ULT.

## Comprehensiveness of reporting

There is significant scope for improving reporting of EHR-based research to facilitate reproducibility and understanding of bias and representativeness. For example through reporting of the timeframe and data linkages in the abstract; sharing code-lists (e.g. in supplemental material or publically available repositories); adequately describing definitions, validation and linkage; and reporting the cohort size during each selection stage.

We found reporting differences between studies using liberal and specific gout definitions in CoR. The former more commonly provided code-lists for definitions and the cohort count at each selection stage. The latter reported more on validation, database population and

linkage, which may be explained by a greater understanding of how healthcare provision determines data collection and use of this to develop a more systematic approach to cohort design and study definitions.

Publication of RECORD guidelines in 2015 may account for the observed improvements in CoR though no studies referenced RECORD. The recent shift from full to partial reporting of the data type in the abstract and of discussion of implications of secondary data use may reflect increasing familiarity of researchers with EHR-based research. Declines in reporting of the cohort size during patient selection may reflect increasing use of centralized databases and modalities of data sharing across EHR-based studies generally.

There were difficulties in assessing certain RECORD items, for reasons that will apply to all EHR-based studies. For example, researchers rarely have full access to the EHR database population and no studies reported this (RECORD item 12.1). We therefore scored instead by whether studies described and appropriately referenced the database (profile, coverage or validation studies), which enables consideration of selection bias. The RECORD items are listed by the area of the manuscript (e.g. abstract, methods) in which they should appear. For items listed in the methods and results, we awarded a score if they were reported in either section. RECORD item 6.3 considers a graphical display of the count of individuals in each linkage stage; we adapted this to apply to all studies by considering display of individuals in each selection stage.

## Risk of bias

RoB measures were generally low, especially in recent studies, but the Cochrane items assessed were "non-applicable" in a third of instances (e.g. those related to co-interventions or estimate adjustment) because many EHR-based studies, both in this review and in general, are descriptive.

## EHR-relevant biases

Other items not considered by commonly used assessment tools are pertinent to both conducting and interpreting EHR-based research. Our systematic review highlights the importance of taking into account the impact of chosen definitions on the findings of research. Other factors include adequacy of the dataset to answer the research question (e.g. hospital versus primary care for studying diabetes) and consideration of temporal changes including, diagnostic certainty, code classifications and completeness and accuracy of recording. Researchers should consider changes in guidelines regarding diagnosis and management, payment practices or policymaking that affect clinical practice or EHR utilization, in study design, analysis and interpretation. Data providers should publically detail all steps undertaken to create a dataset, database profiles and results of data quality assessments. This would facilitate reporting, calibration and the capacity to relate findings back to EHRs for personalized interventions. EHR linkage brings opportunity for multi-site research but may bring site-level bias if appropriate statistical methods (e.g. random-effects models) are not used, yet no multicenter study in this review reported using such methods. This is not specifically covered by common RoB tools for cohort studies.

We looked for discussion of unmeasured confounding, selection bias and changing eligibility over time when assessing reporting of the limitations of secondary data use (RECORD 19.1). These are not assessed by Cochrane but the former two are assessed by ROBINS-I and Newcastle-Ottawa tools [17, 60, 61]. Acquisition bias, where events occurring outside the study window (e.g. diagnoses or prescriptions) affect estimates, was not discussed by any study

**Table 4. Commonly missed factors that affect EHR-based research and recommendations for incorporation into CoR and RoB tools.**

| Factor | CoR Recommendation | RoB Recommendation |
|---|---|---|
| Temporal changes in code classification, EHR system, clinical practice, guidelines or policy | Are these reported on in longitudinal studies? | Are these temporal changes appropriately taken into account (e.g. through adjustment) and/or their impact examined through sensitivity analyses in longitudinal studies? |
| EHR data accuracy, adequacy (e.g. detail) and completeness (including missingness) | Are these reported and previous validation studies referenced correctly? | Is the research question and analysis appropriate, given these? |
| Steps applied and assumptions made during data extraction, processing and cleaning | Are these reported or referenced correctly? | Is the research question and analysis appropriate, given these? |
| Site-level bias | | Is this appropriately addressed in multicenter studies? E.g. include site-level in the model |
| Unmeasured confounding, misclassification bias, selection bias, changing eligibility over time | | Are these appropriately addressed or acknowledged? E.g. replication of analysis with different definitions |
| Bias from unequal follow-up duration | | Are longitudinal studies accounting for follow-up duration? E.g. standardization or minimum follow-up requirement, use of survival methods, use of time-variant variables |
| Bias from competing risks | | Are these appropriately addressed in survival analysis? |
| Bias from change in the population structure, e.g. changes in sites providing data in open cohort studies of long duration | Is description of the population structure (size, demographics) reported over time in longitudinal studies? | Are these appropriately addressed in longitudinal studies? |

nor assessed by the tools. Unequal follow-up duration (data window length), competing risks or loss-to-follow-up due to patient- or EHR system- migration can introduce bias but were not specifically considered in the tools or all studies. Table 4 lists these EHR-related factors for incorporation in future CoR and RoB tools.

## Strengths and limitations

This review comprehensively evaluated all aspects of EHR data-use in gout management research (methodology, use and outcomes, reporting and study quality) and the use of common tools for bias risk and reporting assessment. The limitations include restriction to publications in English and the lack of a standardized term for EHR-based research (e.g. studies may only name a source database or allude to "records"). Due to the EHR focus of the review, we adopted a previously published approach to exclude studies not referencing an EHR-based source even though insurance, claims and administrative data could have been EHR-derived [62].

## Conclusions

EHR-based gout studies used varied case-definitions and medication-related methodology which affects the ability to evaluate and compare treatment outcomes. Nevertheless, they consistently reported that ULT is effective, safe and sub-optimally prescribed. Researchers should improve reporting of methods for reproducibility, particularly through provision of code-lists, data preparation steps and coding validation. Adapted CoR and RoB tools are also required for evaluation of EHR-based research.

## Supporting information

**S1 Fig. Frequency of articles by publication year.** The dotted line represents a polynomial regression line.
(PDF)

**S2 Fig. Boxplot of overall CoR scores for studies by publication year.** Horizontal lines are medians and interquartile ranges (25th and 75th percentiles); whiskers' ends indicate the maximum and minimum values at most 1.5 times the interquartile range from the hinge; dark individual dots are outlier values.
(PDF)

**S3 Fig. Boxplot of overall RoB scores for studies by publication year.** Horizontal lines are medians and interquartile ranges (25th and 75th percentiles); whiskers' ends indicate the maximum and minimum values at most 1.5 times the interquartile range from the hinge; dark individual dots are outlier values.
(PDF)

**S4 Fig. Scatterplot of overall RoB scores for studies by cohort size.** The blue line is the smooth local weighted regression line (LOESS curve). The shaded area indicates the 95% confidence interval.
(PDF)

**S1 Table. Preferred Reporting Items for Systematic Reviews and Meta-Analyses (PRISMA) checklist with reference to this review.**
(DOCX)

**S2 Table. Search terms used with synonyms.** MeSH terms are indicated by '+' and a wildcard by '*'.
(PDF)

**S3 Table. Aim of studies included in the review.** References are cited in S1 File.
(PDF)

**S4 Table. Gout medication types in the studies.** NSAID, non-steroidal anti-inflammatory; ULT, urate lowering therapy.
(PDF)

**S5 Table. Frequency of studies with comprehensive reporting on RECORD items and additional relevant items.** RECORD = REporting of studies Conducted using Observational Routinely-collected Data. References are cited in S1 File.
(PDF)

**S1 File. Supplementary information.**
(DOC)

**S1 Chart. Summary of study characteristics, measures of medication use, outcomes and quality per study.** MPR, medication possession ratio; PDC, proportion of days covered; rmANOVA, repeated-measures analysis of variance; SUA, serum urate level; ULT, urate lowering therapy. *All elements were coded except free-text keywords; where studies use multiple definitions, these are separated by ";". References are cited in S1 File.
(XLS)

## Author Contributions

**Conceptualization:** Samantha S. R. Crossfield, Paul Baxter, Philip G. Conaghan, Mar Pujades-Rodriguez.

**Data curation:** Samantha S. R. Crossfield, Lana Yin Hui Lai, Paul Baxter, Mar Pujades-Rodriguez.

**Formal analysis:** Samantha S. R. Crossfield.

**Funding acquisition:** Samantha S. R. Crossfield, Sarah R. Kingsbury, Philip G. Conaghan.

**Investigation:** Samantha S. R. Crossfield, Lana Yin Hui Lai, Sarah R. Kingsbury, Philip G. Conaghan, Mar Pujades-Rodriguez.

**Methodology:** Samantha S. R. Crossfield, Mar Pujades-Rodriguez.

**Project administration:** Samantha S. R. Crossfield.

**Resources:** Samantha S. R. Crossfield.

**Software:** Samantha S. R. Crossfield.

**Supervision:** Sarah R. Kingsbury, Paul Baxter, Owen Johnson, Philip G. Conaghan, Mar Pujades-Rodriguez.

**Validation:** Samantha S. R. Crossfield, Lana Yin Hui Lai, Mar Pujades-Rodriguez.

**Visualization:** Samantha S. R. Crossfield, Philip G. Conaghan, Mar Pujades-Rodriguez.

**Writing – original draft:** Samantha S. R. Crossfield, Philip G. Conaghan, Mar Pujades-Rodriguez.

**Writing – review & editing:** Samantha S. R. Crossfield, Lana Yin Hui Lai, Sarah R. Kingsbury, Paul Baxter, Owen Johnson, Philip G. Conaghan, Mar Pujades-Rodriguez.

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
