## [Decision Letter · Decision Letter 0]

24 Sep 2019

PONE-D-19-22874

Variation in methods, results and reporting in electronic health record-based studies evaluating routine care in gout: A systematic review

PLOS ONE

Dear dr Crossfield,

Thank you for submitting your manuscript to PLOS ONE. After careful consideration, we feel that it has merit but does not fully meet PLOS ONE’s publication criteria as it currently stands. Therefore, we invite you to submit a revised version of the manuscript that addresses the points raised during the review process.

In addition to the comments of the reviewers, please explain the added value of this manuscript to the field.

We would appreciate receiving your revised manuscript by Nov 08 2019 11:59PM. To enhance the reproducibility of your results, we recommend that if applicable you deposit your laboratory protocols in protocols.io, where a protocol can be assigned its own identifier (DOI) such that it can be cited independently in the future. For instructions see: http://journals.plos.org/plosone/s/submission-guidelines#loc-laboratory-protocols

We look forward to receiving your revised manuscript.

Kind regards,

Michael Nurmohamed, MD, PhD

Academic Editor

PLOS ONE

Journal Requirements:

Reviewers' comments:

Reviewer's Responses to Questions

**Comments to the Author**

1. Is the manuscript technically sound, and do the data support the conclusions?

Reviewer #1: Partly

Reviewer #2: Yes

2. Has the statistical analysis been performed appropriately and rigorously? 

Reviewer #1: Yes

Reviewer #2: Yes

3. Have the authors made all data underlying the findings in their manuscript fully available?

Reviewer #1: Yes

Reviewer #2: Yes

4. Is the manuscript presented in an intelligible fashion and written in standard English?

Reviewer #1: Yes

Reviewer #2: Yes

5. Review Comments to the Author

Reviewer #1: Interesting study/SR of gout studies

Major issue:

1-1977 preliminary criteria by Wallace are too old and only able to classify tophaceous patients; One should at least do a study into newer criteria sets such as the 2015 ACR/EULAR criteria and/or Janssens'calculator as indeed making correct diagnoses in crystal induced arthritis is a world wde issue.

2- paper is missing focus of diagnostic or therapeutic issues in world literature on gout, as both are a challenge

Reviewer #2: A very clear and sound study, addressing the variation in methods, results, reporting and bias risks in EHR-based gout studies . Recommendations to improve studies using EHR-systems are described. Therefore the study answers the study questions. I have no additional remarks.

6. PLOS authors have the option to publish the peer review history of their article (what does this mean?). If published, this will include your full peer review and any attached files.

Reviewer #1: No

Reviewer #2: No

---

## [Author Response · Author response to Decision Letter 0]

1 Oct 2019

Thank you to the Reviewers and for the opportunity to clarify our manuscript. We offer the below response to the comments and questions raised and detail the associated amendments made in the manuscript. References are given below to the associated line changes in the ‘Revised Manuscript with Track Changes.doc’ version of the manuscript.

“In addition to the comments of the reviewers, please explain the added value of this manuscript to the field”

The use of electronic health records (EHRs) in care management is becoming ubiquitous in many countries and secondary data use in research is correspondingly increasing. This systematic review provides evidence of the rise in EHR-based publications examining gout medication (Supplementary Figure 1). The added value of our in-depth evaluation of EHR-based studies relies on the fact that, in this rapidly growing and relatively novel research area, there is wide variation in the methods applied to conduct research as well as the quality of reporting and risk of bias and that this variation might affect the study results. This manuscript represents the first comprehensive assessment of this variation and of the impact that the methods chosen might have on the results obtained. This assessment was performed through a systematic review of articles reporting findings from studies based on the analysis of EHRs from patients, which were conducted to investigate the management of a common chronic disease, gout (amendments made to lines 76-96 to clarify this point). For example, we showed that by using a more specific disease definition to ascertain patients with gout, the likelihood of reporting that appropriate gout medication was prescribed to patients was increased.

The systematic review highlights that there is large scope for improvement of reporting findings from EHR-based studies and quantified the wide variation in the comprehensiveness of reporting, with 73% of published studies discussing the limitations of EHR data use for research, and 5% reporting on key steps of EHR data preparation that can affect the findings. The observed gradual increase in the comprehensiveness of reporting over time is discussed with specific recommendations given for further improvements in future EHR-based studies. The review also reveals the limitations derived from applying current risk of bias assessment tools to EHR-based studies and discusses EHR-relevant biases that are not considered in these tools. It lists recommendations for future assessments of the reporting of, and of evaluations of risk of bias for, EHR-based studies (Table 4). These findings and recommendations, while derived from a review of gout pharmacotherapy studies, are applicable more widely to other EHR-based studies of prevalence, treatment, quality of management and patient outcomes (amendments made to lines 356-357, 366, 417-420, 433-445 to reflect this).

Finally, the manuscript highlights strengths and limitations of EHR-based data in evaluating routine clinical management. It details the impact of the disease definition applied and the capacity to determine the effectiveness of medication in real world settings. In gout, EHR-based studies have been able to show the effectiveness and safety of urate-lowering therapy and their persistently suboptimal use across 1977 to 2017 in terms of low initiation, limited up-titration and low adherence (amendments made to lines to 382-387 emphasise this).

Reviewer 1

Comment 1: “1977 preliminary criteria by Wallace are too old and only able to classify tophaceous patients; One should at least do a study into newer criteria sets such as the 2015 ACR/EULAR criteria and/or Janssens’calculator as indeed making correct diagnoses in crystal induced arthritis is a world wide issue.”

We agree with the comment of the reviewer. The reference has accordingly been amended to the 2015 American College of Rheumatology / European League Against Rheumatism (ACR / EULAR) criteria (lines 127-128), which is used by several studies included in the systematic review. A recommendation has been added to advise that study disease definitions should reflect as much as possible (given the limitations related to the use of routinely collected data) contemporary guideline diagnostic criteria (lines 368-370). Temporal changes in both the guideline diagnostic criteria used by clinicians and the degree of diagnostic certainty are relevant biases for longitudinal EHR-based studies to consider and the Discussion section has been expanded accordingly (lines 442-445). The classification term ‘stringent’, used in measuring the specificity of the definition used by studies, has also been amended to ‘specific’ to improve clarity (lines 123 onwards).

Comment 2: “Paper is missing focus of diagnostic or therapeutic issues in world literature on gout, as both are a challenge”

The objective of the manuscript was to examine the variation in methods, results, reporting and risk of bias in EHR-based studies evaluating pharmacological management of gout. Other specific therapeutic and diagnostic issues in the wider gout-based literature were therefore not examined. None of the EHR-based studies in the review performed adjacent diagnostic assessment and so we were not able to extract information pertaining to issues in diagnosis. The review does however highlight in the Discussion section the problem of suboptimal gout medication management that is recognised globally and is also supported by EHR-based studies [e.g. 1, 2] (amendments made to lines 382-389 to reflect this). A related point is that the diagnostic process used in clinical practice has evolved over time; a recommendation to consider this has been added to the Discussion section (lines 442-445).

Reviewer 2

Comment 1 “A very clear and sound study, addressing the variation in methods, results, reporting and bias risks in EHR-based gout studies. Recommendations to improve studies using EHR-systems are described. Therefore the study answers the study questions. I have no additional remarks.”

Thank you for the positive review.

References

1 Kuo CF, Grainge MJ, Mallen C, Zhang W, Doherty M. Rising burden of gout in the UK but continuing suboptimal management: a nationwide population study. Annals of the rheumatic diseases. 2014 Apr;74(4):661-7.

2 Singh JA, Hodges JS, Toscano JP, Asch SM. Quality of care for gout in the US needs improvement. Arthritis and rheumatism. 2007 Jun 15;57(5):822-9.

---

## [Decision Letter · Decision Letter 1]

10 Oct 2019

Variation in methods, results and reporting in electronic health record-based studies evaluating routine care in gout: A systematic review

PONE-D-19-22874R1

Dear Dr. Crossfield,

We are pleased to inform you that your manuscript has been judged scientifically suitable for publication and will be formally accepted for publication once it complies with all outstanding technical requirements.

With kind regards,

Michael Nurmohamed, MD, PhD

Academic Editor

PLOS ONE

Additional Editor Comments (optional):

Reviewers' comments:

Reviewer's Responses to Questions

**Comments to the Author**

1. If the authors have adequately addressed your comments raised in a previous round of review and you feel that this manuscript is now acceptable for publication, you may indicate that here to bypass the “Comments to the Author” section, enter your conflict of interest statement in the “Confidential to Editor” section, and submit your "Accept" recommendation.

Reviewer #1: All comments have been addressed

Reviewer #2: All comments have been addressed

2. Is the manuscript technically sound, and do the data support the conclusions?

Reviewer #1: Yes

Reviewer #2: Yes

3. Has the statistical analysis been performed appropriately and rigorously? 

Reviewer #1: Yes

Reviewer #2: (No Response)

4. Have the authors made all data underlying the findings in their manuscript fully available?

Reviewer #1: Yes

Reviewer #2: Yes

5. Is the manuscript presented in an intelligible fashion and written in standard English?

Reviewer #1: Yes

Reviewer #2: Yes

6. Review Comments to the Author

Reviewer #1: Thanks for the nice evaluative analytical study in order to improve gout care

Paper is ready for publication

Reviewer #2: The manuscript technically sound, and do the data support the conclusions. I have no further questions

7. PLOS authors have the option to publish the peer review history of their article (what does this mean?). If published, this will include your full peer review and any attached files.

Reviewer #1: Yes: Tim L Jansen

Reviewer #2: No

---

## [Editor Report · Acceptance letter]

16 Oct 2019

PONE-D-19-22874R1 

Variation in methods, results and reporting in electronic health record-based studies evaluating routine care in gout: A systematic review 

Dear Dr. Crossfield:

I am pleased to inform you that your manuscript has been deemed suitable for publication in PLOS ONE. Congratulations! Your manuscript is now with our production department. 

With kind regards,

on behalf of

Prof.Dr Michael Nurmohamed 

Academic Editor

PLOS ONE